# Emergent microrobotic oscillators via asymmetry-induced order

Jing Fan Yang [1,6], Thomas A. Berrueta [2,6], Ada M. Brooks[1], Albert Tianxiang Liu [1,3], Ge Zhang[1], David Gonzalez-Medrano[4], Sungyun Yang [1], Volodymyr B. Koman [1], Pavel Chvykov [5], Lexy N. LeMar[1], Marc Z. Miskin [4], Todd D. Murphey [2] & Michael S. Strano [1] ✉

Spontaneous oscillations on the order of several hertz are the drivers of many crucial processes in nature. From bacterial swimming to mammal gaits, converting static energy inputs into slowly oscillating power is key to the autonomy of organisms across scales. However, the fabrication of slow micrometre-scale oscillators remains a major roadblock towards fully-autonomous microrobots. Here, we study a low-frequency oscillator that emerges from a collective of active microparticles at the air-liquid interface of a hydrogen peroxide drop. Their interactions transduce ambient chemical energy into periodic mechanical motion and on-board electrical currents. Surprisingly, these oscillations persist at larger ensemble sizes only when a particle with modified reactivity is added to intentionally break permutation symmetry. We explain such emergent order through the discovery of a thermodynamic mechanism for asymmetry-induced order. The on-board power harvested from the stabilised oscillations enables the use of electronic components, which we demonstrate by cyclically and synchronously driving a microrobotic arm. This work highlights a new strategy for achieving low-frequency oscillations at the microscale, paving the way for future microrobotic autonomy.

The ability to produce low-frequency oscillations is central to the autonomy of living beings, and is essential to key biological processes such as heartbeats, neuron firings, breathing, and locomotion[1–3]. While complex electronics operates at ever-increasing clock rates of many gigahertz, the frequency of many important biological oscillations seldom exceeds 100 Hz. The slow rate of these oscillations stems from a need to be commensurate with both the energy budget and the natural timescales of underlying biological processes, as in the transport of $CO_2$ in plants[4] and in the galloping of horses[5]. Unlike oscillations arising from external periodic forcing[6–9], these self-oscillations emerge spontaneously from the balancing of competing dynamical processes driving systems away from equilibrium[10–12]—a signature of living systems[13].

In artificial microsystems, however, the production of slow self-sufficient self-oscillations is counterintuitively difficult[14,15]. Generating self-sustaining mechanical oscillations at the microscale typically requires the transduction of complex chemical oscillators (e.g., Belousov–Zhabotinsky reaction[16]) into periodic changes to a system's physical configuration[8,17–21]. Alternative mechanisms for producing self-sufficient mechanical oscillations, based on carefully designed dynamic coupling between responsive elastic materials and thermal[12,22], chemical[11,12,23], or moisture stimuli[24], have typically been demonstrated in millimetre-scale (and larger) devices. In contrast, generating slow periodic electrical signals remains prohibitively challenging aboard untethered microscale devices (Supplementary

[1]Department of Chemical Engineering, Massachusetts Institute of Technology, Cambridge, MA, USA. [2]Center for Robotics and Biosystems, Northwestern University, Evanston, IL, USA. [3]Department of Chemical Engineering, University of Michigan, Ann Arbor, MI, USA. [4]Department of Electrical and Systems Engineering, University of Pennsylvania, Philadelphia, PA, USA. [5]Physics of Living Systems, Massachusetts Institute of Technology, Cambridge, MA, USA. [6]These authors contributed equally: Jing Fan Yang and Thomas A. Berrueta. ✉e-mail: strano@mit.edu

Note 3), given the limited downward scalability of capacitors and inductors[25,26], as well as the power and footprint demands of CMOS oscillators, frequency dividers, and energy modules[27-29]. Despite these challenges, recent progress has shown that self-sustaining electrical oscillations can be produced by modulating electrical resistance with mechanical feedback loops in carefully designed devices, presenting a promising mechanism for sub-500 μm electrical self-oscillators[14].

In this work, instead of relying on complex chemistries, integrated electronics, or elaborate mechanical microstructures, we produce robust electromechanical oscillations aboard a collective of deceptively simple microparticles by exploiting the self-organised properties of their far-from-equilibrium dynamics. By breaking the permutation symmetry of a homogeneous particle collective situated at an air–liquid interface, we reliably control their dynamics to realise simultaneous chemomechanical and electrochemical periodic energy transduction. We achieve this by introducing a particle with an enhanced reaction rate, whose stabilizing effect on the system behaviour we analyse through the lens of asymmetry-induced order. In turn, through a simple bimetallic on-board fuel cell design, we transduce the system's self-oscillations into periodic electrical work to power state-of-the-art microrobotic components, without the need for batteries or external sources of energy.

## Results

### Emergent low-frequency oscillation

Figure 1 presents a system of simple microparticles where low-frequency chemomechanical self-oscillations emerge from the coupling of otherwise self-limiting catalytic reactions easily trapped at equilibrium. Figure 1a shows that each of these microparticles, composed of nothing more than a nanometre-thick Pt patch of radius 125 μm microfabricated beneath a polymeric microdisc, generates a gas bubble when placed at the curved air–liquid interface of a $H_2O_2$ drop via

$$H_2O_2 \xrightarrow{Pt} H_2O + \frac{1}{2}O_2. \qquad (1)$$

This well-studied catalytic reaction has been a long-time favourite in both micro-[30-33] and macroscopic robotics[12,34], noted for the fuel's high energy density and simple chemistry[34].

For a single microparticle situated at the interface, the chemical reaction in Fig. 1a is self-limiting as the bubble grows and gradually blocks off the fuel's access to the catalyst. Consequently, the single-particle system reaches its equilibrium state promptly: The microparticle remains motionless for a prolonged time (Fig. 1d, Supplementary Movie 1) and the bubble asymptotically reaches a terminal radius without rupture (Fig. 1c). However, a drastic change occurs when a second identical particle is introduced to the system. Figure 1b shows that as the microparticles enter each other's proximity, the separately-formed gas bubbles merge. The freed-up catalytic surface area then disrupts the self-limiting chemistry, destabilizing the original single-particle steady-state. This allows the merged bubble to grow beyond its threshold, leading to its rupture (Fig. 1e, $t = 3.2$ s). The collapse imparts an impulse onto the microparticles and propels them in opposite directions, at which point the particles are then drawn back towards one another by the restorational forces: First, the radial component of buoyancy, $\mathbf{F}_g$, globally directs the particles towards the apex of the concave air–liquid interface[9]. Second, the local interfacial deformations result in a mutual attractive capillary force $\mathbf{F}_c$, affectionately known as the "Cheerios effect"[35,36]. The combination of this Cheerios effect and catalytic bubble generation has been observed to produce repetitive back-and-forth motion[37,38] in swarms of tubular swimmers[39,40]. All of these factors sum up to a repeatable cycle of mutual approach, contact, bubble merger, and bubble collapse that we

refer to as particle beating (Fig. 1e). The robustness of this self-sustained cycle is evidenced by the tracked coordinates of the two particles over a course of 280 s (Fig. 1f and Supplementary Movie 2), which contrast the single particle scenario where practically no motion was observed. Notably, while the central challenge in self-oscillatory systems is to keep them away from equilibria[11,15], such states are virtually eliminated from our system by the effectively instantaneous nature of bubble collapse.

We monitored the oscillatory behaviour of the system by tracking its breathing radius $r(t)$ over time, defined as

$$r(t) = \frac{1}{N} \sum_{i=1}^{N} \sqrt{(x_i(t) - \bar{x})^2 + (y_i(t) - \bar{y})^2} \qquad (2)$$

for a collection of $N$ particles each with coordinate $(x_i(t), y_i(t))$ at time $t$. In other words, $r(t)$ is the Euclidean distance from the collective's centroid $(\bar{x}, \bar{y})$ to each particle, averaged over all particles (see annotations in Fig. 1e). The system's periodic beating is evident in the time evolution of $r(t)$ (Fig. 1g, left panel), the limit cycle of its $r(t)$ phase portrait (Fig. 1h, "Methods"), as well as the narrow peak in the recurrence time histogram (Fig. 1i, "Methods"). Taken together, these analyses serve as conclusive evidence of the long-term stability of system oscillations. The analysis in Fig. 1i shows a period of 3.2 s for the two-particle system in 10.7 wt% $H_2O_2$, consistent with Fig. 1g and Supplementary Movie 2. The period remains constant throughout as revealed by the moving-window recurrence analyses (Supplementary Fig. 6, "Methods"), since a negligible 0.02% of the fuel is consumed over 280 s based on stoichiometry. Furthermore, the oscillation amplitude and periodicity are shown to be resilient towards various forms of perturbations (Supplementary Fig. 9). We developed a mechanistic model based on calculated $\mathbf{F}_g$, $\mathbf{F}_c$, and the non-Stokesian drag force $\mathbf{F}_d$ (Supplementary Note 1), and found that it captured even the detailed dynamics of the breathing radius' time evolution (Fig. 1g right panel, also Supplementary Fig. 5). We verified the consistency of the beating frequency across eight sets of independent experiments with 10.7 wt% $H_2O_2$ in Fig. 1j. Additionally, the beating frequency's dependence on $H_2O_2$ concentrations points to a mechanism for exerting fine control over the beating frequency, as predicted by our mechanistic model based on a Langmuir–Hinshelwood kinetics of the catalytic surface (Fig. 1j)[41,42]. In Supplementary Figs. 7, 8, we further explored the dependence of the oscillation amplitude and frequency on $H_2O_2$ volume and particle size. Of note, the stable emergent self-oscillation presented in Fig. 1 does scale down to 250- and 100-μm-diameter particles.

### Persistent periodicity via symmetry-breaking

Our findings in Figs. 2 and 3 show that the stable emergent self-oscillation can be extended well beyond $N = 2$, although curiously only when the system's permutation symmetry is broken and not in a homogeneous system of identical particles. We extracted the bubble burst interarrival time statistics by tracking the time that transpires between each pair of consecutive bursts in recorded experiments (Fig. 2a). In homogeneous systems of identical particles (Fig. 2b), we show that the likelihood of periodic beating dwindles gradually with rising particle counts $N$, reflected in the progressive decay in the sharpness and amplitude of the initial 3.2 s peak corresponding to periodic beating. The decay of collective periodicity is accompanied by an increase in the probability mass of frequent and unpredictable bubble bursts taking place less than a second from one another—a result of bubble mergers and collapses among subsets of particles (see representative $N = 5$ and 8 micrographs in Fig. 2b). Interestingly, we find that the interarrival time distributions of systems beyond $N = 7$ become statistically indistinguishable from those of a Poisson process (Fig. 2b, bottom panel)[43]. This shows that our system's phenomenology can remarkably vary from coordinated and reliable periodic

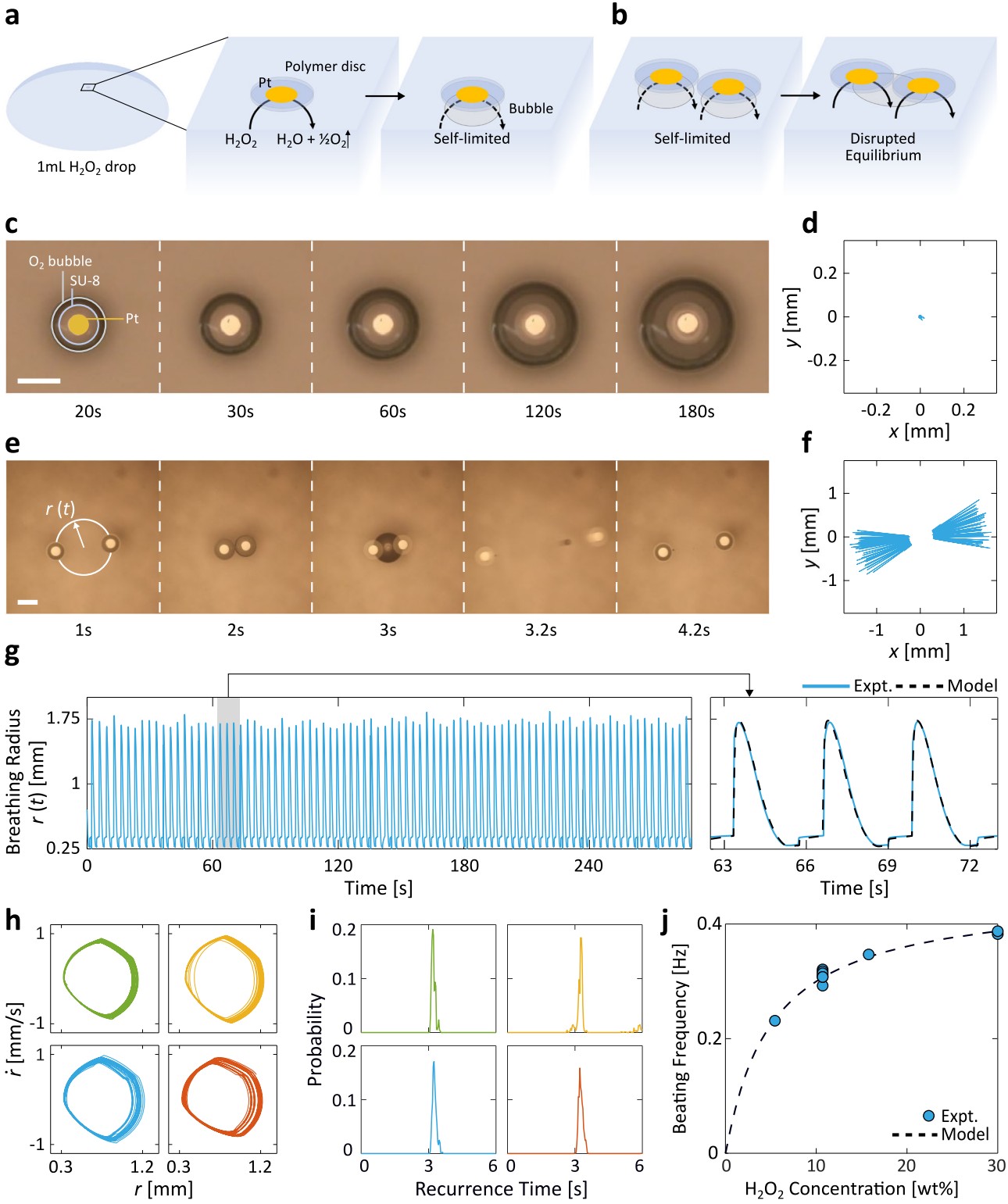

beating to independent and effectively stochastic bubble bursts merely as a function of *N*. The breathing radius trajectory in Fig. 2c confirms the loss of periodicity, as no structure can be discerned from the noisy low-amplitude fluctuations.

The gradual transition towards aperiodicity in Fig. 2b, c points to the nominal fragility of periodic beating as the system size increases. Reasoning that the deliberate introduction of heterogeneity has been shown to produce asymmetry-induced order[44] in complex networked systems[45–47], we investigated the effect of permutation symmetry-

breaking on the robustness of particle beating across system sizes. Based on the role buoyancy plays in the beating physics (Supplementary Note 1), we hypothesised that particles could be made dynamically distinct from one another by controlling the relative size of their accompanying bubble. We tested the impact of this heterogeneity on collective order with Rattling Theory[48,49]. This thermodynamic theory explains the way in which correlations among driven degrees of freedom give rise to system-level fluctuations that govern the long-term stability of system configurations. The magnitude of

**Fig. 1 | Emergence of chemomechanical microparticle self-oscillation.**
**a** Schematic of a self-limited system of a single particle resting still at the air–liquid interface of a $H_2O_2$ drop. The particle is composed of a catalytic patch of Pt (yellow) underneath a polymeric disc (blue). The $O_2$ formation slows down asymptotically over time as the gas bubble restricts the available catalytic surface area. **b** A 2-particle system, in contrast, exhibits an emergent and self-sustained beating behaviour as the bubble merger restores the previously hindered reactivity, thus disrupting the equilibrium state. **c, d** Micrograph sequence (**c**) and tracked particle coordinates (**d**) of a 1-particle system that remains still for an extended period of time. **e, f** Micrograph sequence (**e**) and tracked coordinates (**f**) of a 2-particle system with emergent beating. The breathing radius, $r(t)$, is the distance from the collective's centroid to each particle, averaged over all particles. **g** The long-term breathing radius trajectory of the same system as in **e** and **f** demonstrates the robustness of the beating behaviour. The shaded portion is magnified in the right panel, where the mechanistic model simulations (black, Supplementary Note 1) are shown to match the experimental curve (blue). **h** The phase portraits of 4 independent 2-particle experiments demonstrate reproducible limit cycles with closed-loop orbits, confirming the periodicity of collective beating. Note that to calculate the phase portraits the system's bubble-driven discontinuities were processed through a standard finite-impulse response filter (see Methods). All phase portraits share the same axes. **i** The recurrence histograms of the same 4 experiments all display a narrow peak centred at a period of 3.2 s, consistent with visual evidence in **e**. All histograms share the same axes. **j** The beating frequency can be tuned with the concentration of $H_2O_2$. The dependence predicted by the mechanistic simulations, on the basis of a Langmuir–Hinshelwood kinetics (black curve), matches the experimental measurements (blue markers). Scale bars, 500 µm.

these fluctuations, as quantified by Rattling $\mathcal{R}$, serves as an index describing the system's degree of disorder. Since lowering $\mathcal{R}$ requires substantial correlations among degrees of freedom, systems in low-$\mathcal{R}$ configurations often exhibit emergent order.

We constructed a theoretical model that analytically connects a bubble's relative size with its contribution to system-level fluctuations, and in turn collective order (Supplementary Note 2). The model's predictions in Fig. 2d suggest that any deviation in a single particle's bubble size relative to the rest of the ensemble (i.e., with relative burst intensity away from 1x) results in a more orderly system as quantified by lower $\mathcal{R}$. Interestingly, the reduction in $\mathcal{R}$ is found to be particularly significant when a bubble larger (and stronger) than its peers is introduced, which we confirmed with experiments. We note that this novel mechanism for asymmetry-induced order applies to a broad class of complex systems wherein parametric heterogeneities control the fluctuations of strongly interacting elements (see Supplementary Note 2).

We broke the permutation-symmetry of the original system experimentally by adding a "designated leader" (DL) particle with an enlarged Pt patch of radius 175 µm (Fig. 2e). Note that since the nanometre-scale thickness of the Pt layer is negligible compared to that of the unchanged 10-µm-thick polymeric microdisc, the DL design does not alter the particle's volumetric geometry. However, the heterogeneity among the catalytic surface areas translates directly to unequal bubble growth rates between the DL and its neighbours, which in turn drastically affects their collective dynamics in accordance with our theoretical predictions in Fig. 2d: We observe robustly periodic bubble collapses across $N$ in the sharp peaks of the interarrival distributions in Fig. 2e, suggesting that DLs are able to sustain the periodicity of particle beating even at high particle counts. Figure 2f depicts the time evolution of the breathing radius for a system of $N = 7 + 1\text{DL}$ particles (see also Supplementary Movie 4). In contrast to the homogeneous $N = 8$ system (Fig. 2c), the heterogeneous DL system exhibits a stable long-term self-oscillation with a period of 14.2 s, owing to the broken permutation symmetry.

Figure 3a(i–vii), b(i–vii) explains the microscale physics arising from the intentionally broken symmetry (see also Supplementary Movie 3). When a DL particle with an enlarged Pt patch is paired with a non-DL particle, the heterogeneity in bubble sizes leads to the subsumption of the non-DL particle bubble into the DL bubble upon contact (Fig. 3a(ii–v) and b(ii–v)). This coalescence behaviour is distinct from that of equal-sized bubbles previously shown in Fig. 1b, where an unstable merged bubble forms halfway between the particles. Instead, the merged bubble sticks to the former location of the large parent bubble underneath the DL particle, seen in Fig. 3a(iii) and (v). This behaviour falls under the sticking bubble regime in the literature, a phenomenon long observed in experiments[50,51] but only recently thoroughly studied and theorised in a catalytic $H_2O_2$ bubble system[52]. Importantly, contrary to the more intuitive moving bubble regime where the merged bubble sits at the centre of mass of its parents[53,54], the coalescence behaviour transitions into the sticking

regime only as the parent bubbles differ sufficiently in size[52], or, in other words, with sufficient particle heterogeneity. As shown in the rest of Fig. 3a, b, the two particles in the system undergo several rounds of small-scale bubble coalescence, eventually causing the DL bubble to collapse. We find that the bubble's rupture radius is approximately 1.7 times larger than that for a homogeneous system shown in Fig. 1f, stabilised by the particle sitting directly on top. This contributes to an even lower-frequency chemomechanical oscillation (Figs. 2f and 3f) than that previously observed in homogeneous systems (Figs. 1i and 2b).

Figure 3c, d contrast the breathing radius phase portraits between homogeneous and heterogeneous systems of different $N$. We observe that the homogeneous systems experience a decay of periodicity evidenced by the gradual collapse of limit cycle orbits in its phase portraits as a function of $N$, consistent with trends in Fig. 2b. In contrast, the heterogeneous systems' limit cycles are robust to variations in $N$, retaining their closed-loop phase-space orbits. To rigorously quantify the effect that DLs have on collective periodicity, we analysed the recurrence structure of the dynamical trajectories across system sizes (see "Methods")[55]. As sketched in Fig. 3e, recurrence analyses capture the dynamical properties of system behaviours by measuring the time the system takes to return to a given state's neighbourhood. The set of all such time intervals is compiled into a recurrence histogram (Fig. 3f) whose recurrence entropy can be used to quantify the complexity of dynamical trajectories[56], with perfect periodicity corresponding to zero entropy.

The linear entropy increase for homogeneous systems as a function of $N$ (Fig. 3g) corresponds to the increasing disorder in the system's recurrences that is consistent with the progressive loss of periodicity observed in Figs. 2c and 3c. Also in accordance with earlier qualitative trends in Figs. 2f and 3d, the recurrence entropy of the DL system is locally invariant to changes in $N$, thereby providing quantitative evidence of the robustness of the periodic beating induced via symmetry-breaking. While we find that the system's invariance to particle number holds up to $N = 11$, we leave the study of larger particle systems for future work (Supplementary Figs. 10, 11).

## Self-oscillating microgenerators

Through a simple modification to the particle design, we are able to harness the robust chemomechanical beating to generate an oscillatory electric signal. As illustrated in Fig. 4a, b, we fabricated particles with a Pt pattern closely lined up with (though spatially separate from) an additional metal patch of either Au or Ru (see Methods). With the bimetallic design, the previously auto-redox catalytic decomposition of $H_2O_2$ on Pt is in part separated into an oxidation half-reaction on Pt and a reduction half-reaction on Ru (Au)[30,31,57]:

$$\begin{aligned} \text{Pt} &: H_2O_2 \rightarrow O_2 + 2H^+ + 2e^- \\ \text{Ru (Au)} &: H_2O_2 + 2H^+ + 2e^- \rightarrow 2H_2O \\ \text{Overall} &: 2H_2O_2 \rightarrow 2H_2O + O_2. \end{aligned} \tag{3}$$

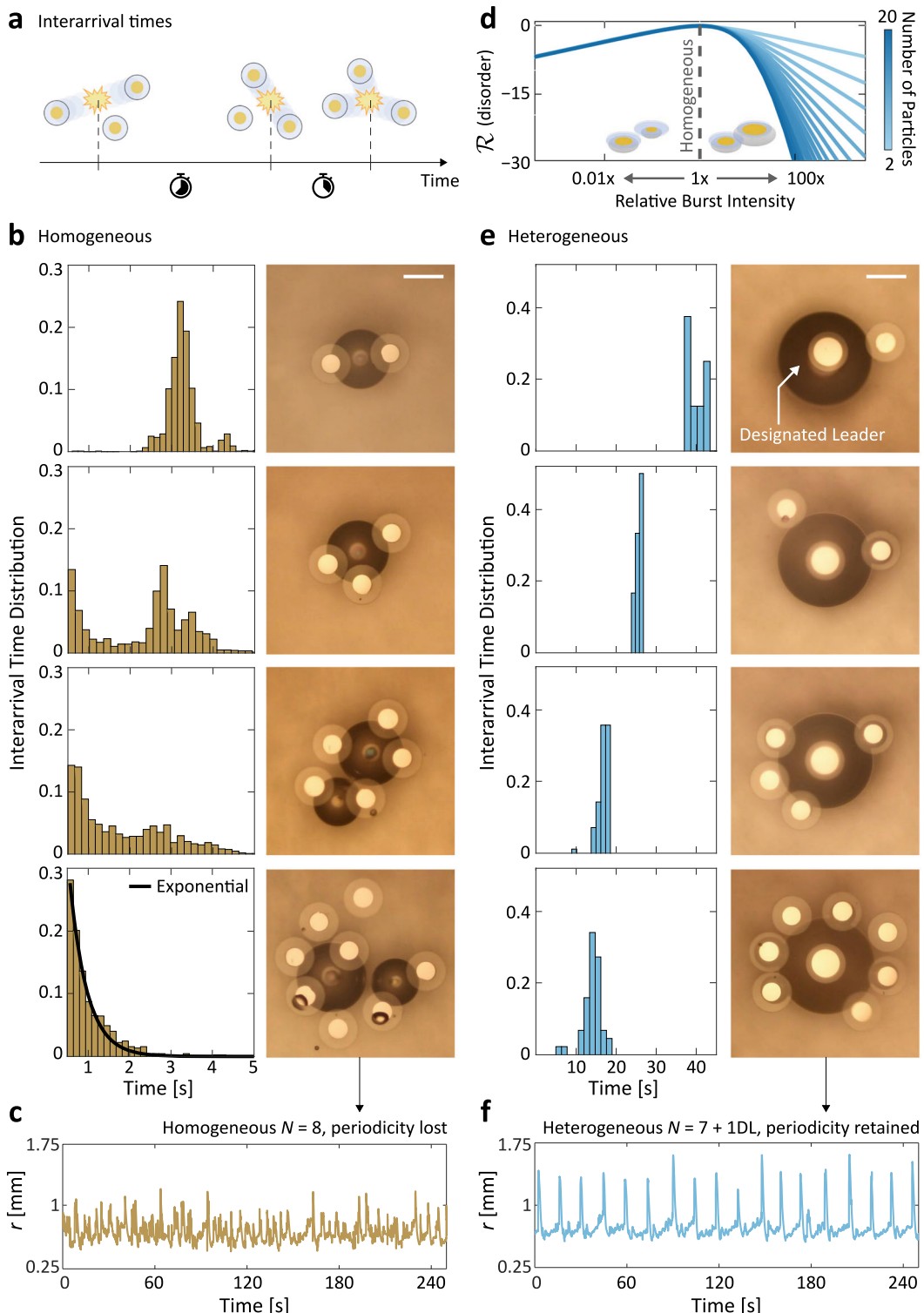

**Fig. 2 | Observations of emergent order via symmetry-breaking. a** Schematic of interarrival times in a system of beating microparticles, defined as the time that transpires between two consecutive bubble collapses. The interarrival time distribution should be tight (i.e., a single peak) in a perfectly periodic system, and broad in an aperiodic system. **b** (top to bottom) Interarrival time distributions and optical micrographs for homogeneous systems of $N = 2$, 3, 5, and 8 identical particles. As $N$ increases, the collective system periodicity gradually decays and transitions to an exponential interarrival distribution at $N = 8$ (bottom, black curve). Scale bar, 500 μm. **c** Indeed, we observe that the breathing radius of a homogeneous $N = 8$ system is not periodic. **d** Asymmetry-induced order across $N$ predicted by Rattling Theory. A quantification of collective disorder, the system's Rattling $\mathcal{R}$ is predicted to be lower (i.e. more orderly) if the relative burst intensity of one particle is increased beyond or decreased below 1x, which signifies homogeneity. This is experimentally realised by modulating the Pt patch size on a "designated leader" (DL) particle relative to the others. The curves are offset to make all $\mathcal{R} = 0$ at 1x intensity to highlight the effect of system heterogeneity on Rattling. See Supplementary Note 2 for a detailed discussion of the analytical model. **e** Same as (**b**), but for heterogeneous systems of equal particle numbers, where the DL broke the permutation symmetry. In contrast to the homogeneous systems (**b**), they remain robustly periodic across $N$. It is important to recognise that the polymeric disc size of a DL is unchanged. Scale bar, 500 μm. **f**, Breathing radius for an 8-particle DL system (i.e., $N = 7 + 1$DL), which reliably beats periodically. The period of 14.2s extracted from $r(t)$ coincides with the most probable interarrival time in **e** (bottom).

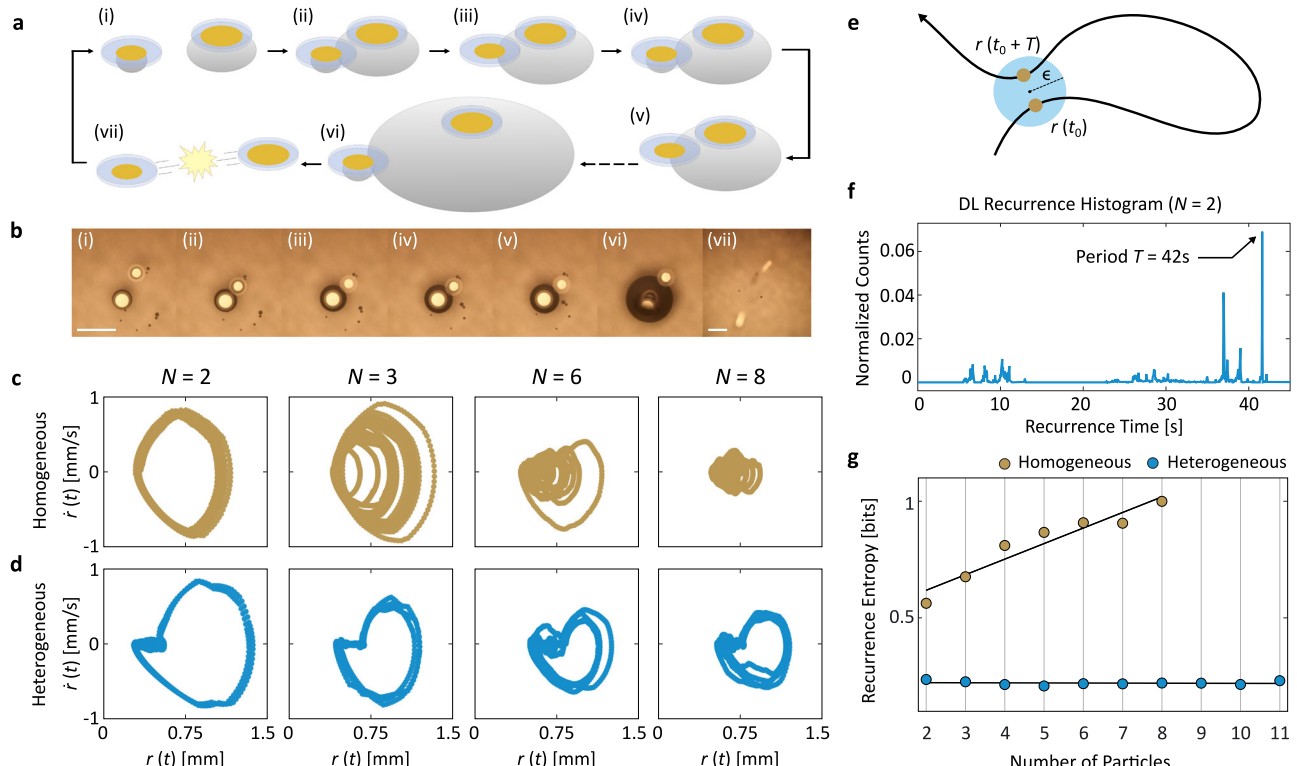

**Fig. 3 | Designated leaders induce periodic limit cycles. a, b** Features of DL beating explained with schematic (**a**) and micrograph sequence (**b**) of a 2-particle heterogeneous system. The leader particle is able to grow a large bubble promptly and subsume the smaller bubbles of neighbouring particles across several rounds of bubble coalescence. Scale bars, 1 mm. **c, d**, Phase portraits of homogeneous (**c**) and heterogeneous (**d**) systems of $N = 2, 3, 6,$ and 8. Only the latter is able to maintain the closed-loop orbits at high particle counts. **e** Schematic of recurrence time calculation. The recurrence time is the time it takes to return from a given system configuration to the neighbourhood of said configuration (see "Methods"). **f** Recurrence histogram compiling all of the recurrence times observed across experiments of the 2-particle heterogeneous system ($N = 1 + 1$DL). **g** Recurrence entropy as a function of $N$ for both homogeneous (yellow) and heterogeneous/DL (blue) systems. Low recurrence entropy is a quantitative indicator of periodic behaviour. The homogeneous system's recurrence entropy trends upward, suggesting a decay in periodicity, while the DL system's entropy remains low in accordance with its observed periodicity even at high $N$.

Consequently, a potential difference is established at the two electrodes that essentially transforms the particle into an on-board fuel cell. These same principles have been previously used to generate voltages in nanomotors, where bimetallic rods and nanoparticles are propelled electrokinetically by the accompanying electric field[58–60]. A micrograph of our fabricated prototype is displayed in Fig. 4b. Note that the metallic leads extending outwards were added to facilitate electrical characterization of the devices and are not necessary to their operation. The leads were passivated and hence do not participate in any electrochemical reactions. The Pt-Ru and Pt-Au fuel cell devices measured open-circuit voltages of 144.9 mV ± 2.4 and 21.4 mV ± 3.5, respectively, in a 25.8 wt% $H_2O_2$ solution with 0.075 M $KNO_3$ added for conductivity (see "Methods" and Supplementary Fig. 13). These values are in line with prior mechanistic studies[30,31] (Supplementary Note 4). Under the same conditions, the Pt–Ru fuel cell delivers a short-circuit current density of 1.71 mA/$cm^2$ ± 0.38 and a current of 56.7 nA ± 12.4. As a benchmark, a significantly larger 1.5 × 6 cm thermo-mechano-electrical self-oscillator reported recently recorded a peak current of ~47 nA[61]. The dependence of the current density on $H_2O_2$ concentration is summarised in Fig. 4c (also Supplementary Fig. 14).

As before, the system's collective beating drives the synchronised formation and collapse of bubbles on each particle. However, unlike previous experiments, here the instantaneous size of the bubble also modulates the electrical conductance from one electrode to the other (Fig. 4a, $N = 2$ for demonstration). This effect, in conjunction with the fuel cell's voltage, enables the on-board generation of oscillatory currents that are in phase with the mechanical

beating (Supplementary Fig. 16). In a Pt-Ru device, we observe that the ON/OFF ratio between maximal and minimal currents can exceed $10^6$, corresponding to when the bubble is absent and at its threshold size. Importantly, the same chemical energy harnessed from the environment is used to simultaneously drive the mechanical oscillation, generate the electrical voltage, and modulate the electrical conductance. Multifunctionality of this kind is emblematic of emerging paradigms such as embodied energy[62], and is crucial to the development of efficient microsystems.

Figure 4d, e exemplifies the beating system's capability to cyclically drive a microrobotic load with its self-generated oscillatory electrical current. In this proof-of-concept demonstration, we wired the Ru electrode of a fuel cell particle to a state-of-the-art Pt–Ti electrochemical microactuator (see Fig. 4d and "Methods"), originally invented for a tethered sub-100 μm walker[63]. In our experimental configuration, charged species from the electrolyte is desorbed from the Pt surface of the bimorph microactuator as current passes through, causing it to deswell and its curvature/length to change. Evident in Fig. 4e, the periodic actuation of the bimorph (red curve, representative snapshots in Fig. 4d, also Supplementary Movie 5) is driven by the periodic spikes in the current signal (blue curve), which in turn is modulated by the chemomechanical beating of two particles. Because the outer radius of the Pt electrode (Fig. 4b) exceeds the 125 μm patch radius of a standard particle, the system is stabilised by the added heterogeneity, which also explains the observed sub-0.03 Hz beating frequency. In contrast, the control experiments in Fig. 4e show the actuator idling in the absence of a second particle and hence any mechanical beating. By harnessing the emergent power generation of

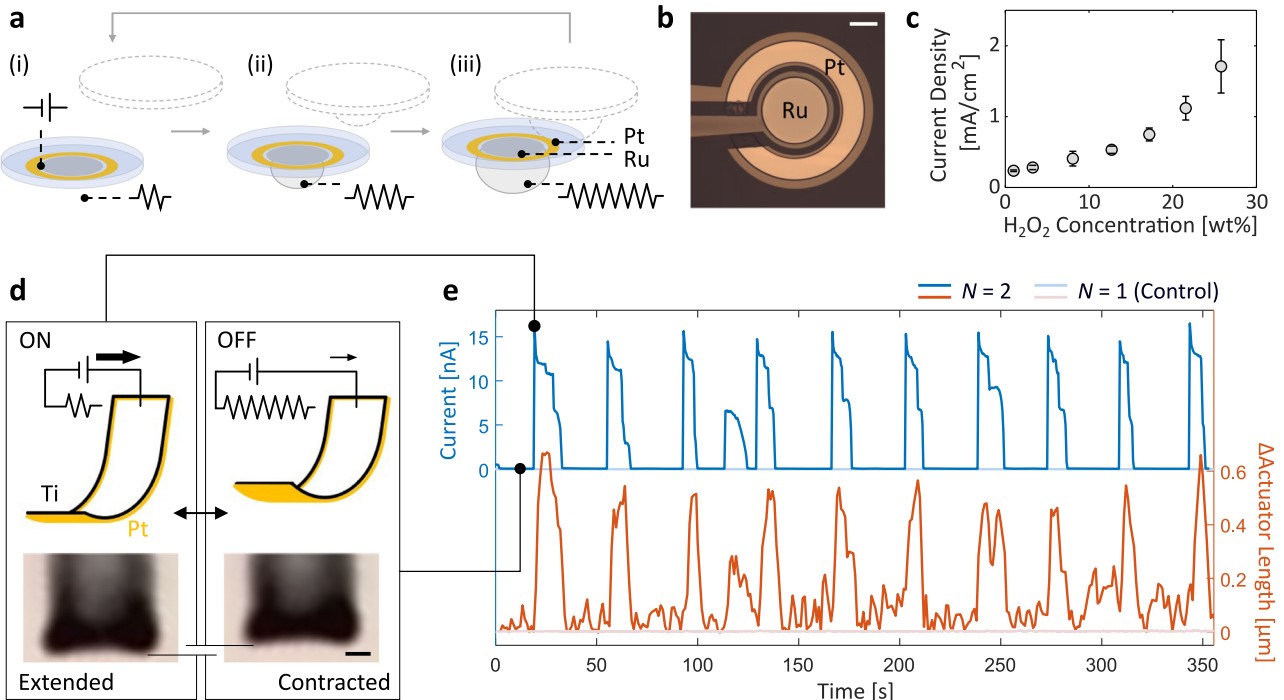

**Fig. 4 | Self-organised oscillation powers a microrobotic arm. a** Schematics of the generation of an oscillatory electrical current from chemomechanical beating. The pair of metals (Pt–Ru or Pt–Au) patterned on a polymer base constitute the electrodes of a $H_2O_2$ fuel cell, which serves as an on-board voltage source. The periodic bubble growth and collapse in a beating system separately modulates the electrical resistance between the electrodes, leading to an oscillatory current. **b** Optical micrograph of a typical Pt-Ru fuel cell particle. The entire surface, less the electrode area, is passivated with a thin layer of insulating SU-8 polymer (shaded). The metallic leads on the left are not necessary for device operation and are included to facilitate measurement. Scale bar, 100 μm. **c** Short-circuit current density as a function of $H_2O_2$ concentration for a Pt-Ru device. **d, e** Cyclic motion of a microrobotic actuator driven by the oscillatory current. The schematics and micrographs in **d** show the extended and contracted states of the actuator respectively under the ON and OFF current conditions, as modulated by the bubble size. The current measurement over time and the actuator length change (**e**) closely match, confirming that the cyclic actuation is driven by the oscillatory current, which itself is emergent from the particle beating. Scale bar, 2 μm.

an ensemble of microparticles, we have demonstrated the design and modular interoperability of key microrobotic components—energy sources and locomotive elements—based on the physics of self-organization.

## Discussion

Through the discovery of physical mechanisms for asymmetry-induced order, we constructed self-oscillating electrical generators capable of powering on-board microrobotic components from the interactions of simple microparticles. Our results stand in contrast to more traditional microrobotic approaches focusing on the design of intricate electromechanical assemblies to produce alternating electrical currents[14]. By relying on our system's self-organised behaviours, we circumvented the design of complex contraptions to harvest and transduce chemical energy into periodic electrical and mechanical work—a crucial step towards fully-autonomous microrobots[62,64]. The use of on-board electrical currents will enable the integration of sensors and computational elements to enrich physical microparticle interactions[65], forming the basis for future collectives wherein the long-envisioned potential of complex inter-particle communications can be implemented[40]. We plan on extending our approach into studying larger collections of microparticles in search of general principles for the top-down design of active matter systems, where an understanding of system symmetries and environmental forcing may enhance their task-capability. Unifying perspectives from their respective fields, our work suggests that future microrobots and active matter systems may become more robust and task-capable when we design them to exploit the physics of the environments they inhabit.

## Methods

### Fabrication and liftoff of microparticles

The fabrication process is summarised in Supplementary Fig. 4. SU-8 2010 photoresist was spun on a Si wafer at 3000 rpm for 1 min. It was baked at 65 °C for 1 min and 95 °C for 2 min. The SU-8 discs were defined by exposure with a Karl Süss MA6 Mask Aligner at a dose of 140 mJ (365 nm). The wafer was baked post-exposure at 65 °C and 95 °C, respectively, for 1 and 2.5 min. The resist was developed in SU-8 developer for 2.5 min, soaked in isopropanol, and blow dried. The wafer was optionally hard baked at 115 to 180 °C for 10 min to 2 h.

LOR 3A photoresist was spun onto the sample at 1000 rpm for 1 min. This was optionally followed by a second spinning step at 2000 rpm for 30 s to ensure that the coating was uniform at the periphery. The sample was baked at 180 °C for 4 min. Shipley S1818 photoresist was spun at 2000 rpm for 1 min and subsequently heated at 115 °C for 1 min. The LOR and Shipley parameters were optimised to ensure a full coverage over the 10-μm-thick SU-8 discs. The sample was aligned and exposed at 140 mJ (405 nm). It was then developed in AZ 726 MIF for 1.5 min. The sample was washed with running DI water and blow dried.

The Pt metal patches were deposited with a Denton e-Beam Evaporator. A typical patch consists of 5 nm of Cr or Ti adhesion layer and 50 nm of Pt. The photoresists were stripped in Remover PG.

The fabricated microparticles were lifted off the wafer substrate in 45 °C 1M KOH solution, which etched away Si (Supplementary Fig. 4b). The process typically took 30 to 50 min. The microparticles were collected by a transfer pipette and then washed repeatedly with DI water until the solution's pH was neutral. Alternatively, the microparticles were first coated with PMMA A4 (polymethyl methacrylate) before being lifted off in 90 °C 1M KOH solution (Supplementary

Fig. 4c). The microparticle array on the PMMA sheet was picked up by a clean piece of wafer. The PMMA was carefully dissolved away with acetone and the particles were washed by and stored in DI water.

## Experimental characterization of beating behaviour

In a typical experiment, 1 mL of $H_2O_2$ solution (10.7% unless otherwise noted, VWR International, LLC, Radnor, PA) is dispensed gently onto a polystyrene Petri dish (VWR International, LLC, Radnor, PA). Two methods were used to transfer the micro-oscillators from their vial to the $H_2O_2$ droplet. In the "wet" method, they could be collected with a narrow-tipped transfer pipette along with a small amount of water, and subsequently transferred onto the droplet. The introduction of a minor amount of diluent as well as the occasional need to flip over a particle can be avoided with an alternative "dry" process. First, a particle was wet transferred onto a glass slide with a transfer pipette. Excess water was carefully wiped off while the particle was not completely dried. A drop of $H_2O_2$ solution was then added. This step allowed the operator to check the orientation of the particle on the glass slide prior to its transfer to the droplet. A quartz NMR sample tube was used to directly pick up the particle dry, a process assisted by surface tension. Note that the other end of the tube was, of course, capped. Lastly, the dry particle with the correct orientation was gently placed atop the 1 mL $H_2O_2$ droplet under the camera.

The beating behaviour was recorded as 30 fps videos with a Canon Rebel T6i camera (Canon U.S.A., Inc., Huntington, NY). The optical system comprised a magnification lens (MVL12X20L), a coaxially focusable zoom lens (MVL12X3Z), and an extension tube (MVL12X3Z), all purchased from Thorlabs, Inc., Newton, NJ. The setup followed that described previously in ref. 66. The illumination source was a MI-150 Fiber Optic Illuminator from Edmund Optics Inc., Barrington, NJ.

## Phase and recurrence analyses of particle beating

The recorded videos of the beating systems were processed with the Image Processing Toolbox of MATLAB (MathWorks, Inc., Natick, MA). The particle centres were identified from each frame of the videos with the standard `imfindcircles` function, a circle-finding algorithm based on circular Hough transform[67]. Given a collection of particle trajectories from an experimental trial, the main observable from which to construct the phase portraits shown in Fig. 3 was the breathing radius $r(t)$ as defined in Eq. (2). The phase portraits were then constructed by plotting the coordinates of $v(t) = [\dot{r}(t), r(t)]$ after applying a low-pass filter, and the time-derivative of the breathing radius was estimated via finite differencing.

Equipped with the dynamical observables defined above, the recurrence properties of a system can be analysed by finding how often and how quickly the system returns to a neighbourhood of $v(t)$. Hence, for a given experiment comprised of $K$ samples we collect data at times $t_i = i\Delta t, \forall i \in \{0, \cdots, K-1\}$ with sampling rate $\Delta t$. While in principle this is all one needs in order to quantify recurrence statistics[55], an additional step must be taken in order make the calculation robust. We augmented our $v(t_i)$ vectors by "embedding" the time-series according to an integer parameter $m$[56]. This resulted in a modified set of coordinates, $v_m(t_i) = [v(t_i), \cdots, v(t_{i+m-1})]^T$, from which to robustly calculate our recurrence statistics. Finally, to derive the recurrence properties of a system from an experimental dataset we calculated its recurrence set

$$R_s = \{|t_i - t_j| : \ ||v_m(t_i) - v_m(t_j)|| < \epsilon, \forall i, j\}, \tag{4}$$

over all valid indices. Note that $m$ and $\epsilon$ are a fixed choice of positive non-zero embedding dimension and neighbourhood size parameters, respectively. With this set now defined, we could calculate a recurrence histogram from the set $R_s$ using any standard scientific computing package, as in Fig. 3. Additionally, we note that the histogram can be normalised into a pseudo-probability distribution that expresses the

likelihood $p(T)$ that a system exhibits a recurrence after $T$ seconds. The dominant frequencies plotted in Figs. 1, 3 and Supplementary Fig. 6 were computed from the $T$ of maximum likelihood from the corresponding recurrence analyses.

As we are interested in characterizing the onset of periodicity across collectives of beating particles, we must construct a measure capable of differentiating the diversity of behaviours we observed. For this purpose, we made use of the entropy of the recurrence probability distributions. As an example, consider a system with a single perfectly oscillatory mode. Then, its recurrence distribution would be a delta function corresponding to its period of oscillation, and thus have zero entropy. If one were to introduce noise or uncertainty into that single oscillatory mode, then probability mass would spread around the delta peak and generate non-zero entropy. Likewise, if the system were to have multi-modal (but deterministic) oscillation, probability mass is now shared between the peaks of the distribution, leading to non-zero entropy.

As the behaviour of a system becomes increasingly complex, it has been shown that the recurrence distribution entropy is a useful metric to quantify this shift that has known connections to both Kolmogorov-Sinai and Rényi entropies[68], as well as the correlation sum in chaos theory[69]. However, in order to compare the recurrence entropies of systems with different magnitude- and timescales, we first normalised our data in two ways. First, we applied min-max normalization to the coordinates of $p(T)$, which allows one to use the same $\epsilon$ in the calculation of the recurrence set. Second, we normalised the elements of $R_s$ according to its maximum (while keeping the number of histogram bins constant across systems) in order to study the structure of system recurrences without confounding variables. The result of this process can be seen in Fig. 3.

## Fuel cell fabrication

LOR 20B photoresist was spun onto a Schott Borofloat 33 wafer (UniversityWafer, Inc., Boston, MA) at 3000 rpm for 1 min and baked at 180 °C for 4 min. Shipley S1805 photoresist was spun at 3000 rpm for 1 min and baked at 115 °C for 1 min. The sample was exposed at 82.5 mJ (405 nm). It was then developed in Microposit MF-319 developer for 65 s. The sample was washed with running DI water and blow dried. A Denton e-Beam Evaporator was used to deposit 10 nm of Ti and 50 to 100 nm of Pt. The photoresists were stripped in Remover PG. For the deposition of a second metal, be it Au or Ru, LOR and Shipley resists were spun, baked, exposed, and developed the same as described above. 10 nm of Ti and 100 nm of Au was deposited with an electron beam evaporator. Alternatively, 50 nm of Ru was deposited as the deposition was slow. The photoresists were stripped in Remover PG.

The SU-8, LOR, and Shipley resists were all purchased from Kayaku Advanced Materials, Inc., Westborough, MA, in addition to the SU-8 developer, MF-319 developer, Remover PG, and PMMA. The AZ 726 MIF was purchased from MicroChemicals GmbH, Ulm, Germany.

Finally, a passivation layer of SU-8 was defined on top of the metal electrodes. For the convenience of the electrical measurements that followed, the SU-8 were patterned as either 5mm-by-5mm or 11mm-by-11mm square islands with the active electrode area at the centre exposed. SU-8 2002 was used but the precise thickness was inconsequential.

## Fabrication and characterization of microactuators

The Pt–Ti bimorph microactuators were fabricated on a Cu sacrificial layer at University of Pennsylvania's microfabrication facility according to the procedures previously reported[63]. The actuators were lifted off overnight in a 4 mg/mL ammonium sulfate solution, which etched away the Cu substrate. The actuators were subsequently transferred to a phosphate-buffered saline (PBS) solution.

In Fig. 4 of the main text, the bimorph microactuators were cyclically driven by the oscillatory electrical current signal generated

by the oscillatory beating between a Pt–Ru fuel cell device and a Pt-decorated beating particle. Each microactuator was picked up by a parylene-coated Pt-Ir monopolar electrode (PI2003X.XA3, 0.1MΩ, Microprobes for Life Science, Gaithersburg, MD) in PBS. The parylene coating prevented unnecessary current leakage into the electrolyte. The Pt-Ir electrode connected to the Ru electrode of the fuel cell device via a probe station (Advanced Research Systems, Macungie, PA) and a W probe (The Micromanipulator Company, Carson City, NV). The probe station read out the real-time current with a custom MATLAB code. The Pt electrode of the fuel cell, via a W probe, was connected to a Pt wire partially immersed in the PBS solution.

A 30% $H_2O_2$ solution and a 0.5M $KNO_3$ solution were mixed at a volumetric ratio of 85:15. The salt was included to enhance the electrolyte's electrical conductivity. For the self-oscillation to take place, 8.5 $\mu$L of the prepared mixture was dropped atop a fuel cell device on the wafer. A beating particle was subsequently transferred to the same solution using the transfer method described earlier. The actuation was recorded with the same optical setup described above mounted over the probe station.

### Actuation analysis of microactuators

The extent of actuation as a function of time was extracted from the recorded videos described in the previous section. A standard canny edge detection algorithm with pixel magnitude thresholds was applied via OpenCV[70]. A boundary representing the outline of the actuator was extracted, which could then be used to define a coordinate system aligned and centred along the long edge of the actuator over the duration of the video—a crucial step for reducing measurement drift. From this coordinate system, the length of the actuator was then simply defined according to the nearest actuator boundary pixels along the vertical axis. Finally, in order to mitigate the effect of fluctuations and mechanical vibrations, a standard low-pass finite-impulse response (FIR) interpolation scheme was applied to the actuator length signal over time[71].

## Data availability

The data supporting the findings of this study are available from the corresponding author upon reasonable request.

## Code availability

The code supporting the findings of this study is available from the corresponding author upon reasonable request.

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

## Acknowledgements

The authors are appreciative of funding from the US Army Research Office MURI grant on Formal Foundations of Algorithmic Matter and Emergent Computation (W911NF-19-1-0233) for the computational and metrological tools as well as the analysis of emergent behaviour used in this work. Funding from the US Department of Energy (DOE), Office of Science, Basic Energy Sciences (grant DE-FG02-08ER46488) supported device and experimental design, fabrication and component engineering. We acknowledge helpful discussions with Dana Randall, Andrea Richa, Daniel Goldman, Jeremy England, Ana Pervan, Annalisa Taylor, Shengkai Li, Hridesh Kedia, Mahesh Kumar, Matthias Kühne, Joy Zeng, Jorg Scholvin, and Kaihao Zhang. Microfabrication for this work was performed at the Harvard University Center for Nanoscale Systems (CNS), a member of the National Nanotechnology Coordinated Infrastructure Network (NNCI), which is supported by the National Science Foundation under NSF award No. ECCS-2025158; the MIT.nano microfabrication facility at Massachusetts Institute of Technology; and University of Pennsylvania. G. Z. acknowledges the support from MathWorks Engineering Fellowship.

## Author contributions

J.F.Y., A.T.L., and M.S.S. conceived the experiments. J.F.Y., A.T.L., G.Z., and S.Y. fabricated the microparticles. J.F.Y. and A.T.L. carried out the collective beating experiments. J.F.Y., T.A.B., A.T.L., A.M.B., and L.N.L. processed the experimental data. T.A.B. performed analyses on the phase portraits, interarrival distributions, and recurrence histograms. T.A.B. derived theoretical results on asymmetry-induced order and Rattling model. P.C. contributed to theoretical analyses. J.F.Y. and A.T.L. studied the physics of the beating mechanism. J.F.Y. constructed the mechanical model and performed the simulations. J.F.Y., A.M.B., and M.S.S. designed the electrochemical fuel cells. J.F.Y. fabricated the fuel cells and performed the actuator experiments with G.Z. D.G.-M., and M.Z.M. fabricated the microactuators. J.F.Y, T.A.B., T.D.M., and M.S.S. wrote the manuscript with all authors contributing.

## Competing interests

The authors declare no competing interests.
