## [Peer Review File · Nature Communications]

Emergent microrobotic oscillators via asymmetry-induced orderREVIEWER COMMENTS

Reviewer #1 (Remarks to the Author):

The author reported an oscillatory microrobot fueled by hydrogen peroxide. The bubbling generation, merging with neighbor bubbles, bubble repulsion and particle attraction lead to a self-sustained oscillation system. The author studied multiple particle systems from experimental and theoretical pathways. The oscillatory motion could be used to trigger alternative electrical signals, which could power periodic motion of a micro-actuator. It is impressive that the authors were able to develop the microgenerator application from the oscillation phenomenon. Meanwhile, this method suffers from many intrinsic limitations (in terms of stability, long-term operation, etc.), though it is understandable for a proof-of-concept application. Overall, it is an interesting story with clear description. The reviewer recommends publishing the article in Nature Communication after addressing some comments below:

1. The author showed the tunability of the oscillation frequency. Is there any way to change amplitude, which is radius, in this system?
2. For the homogeneous system where relative burst intensity is 1, increasing the number of particles leads to loss of periodicity. However, why does increasing the number of particles not change R?
3. What is the energy efficiency of the microgenerator and mechanical oscillation? Will the generation of electrical signals decrease the mechanical oscillation's amplitude?
4. It is suggested to add the radius change over time in Figure 4e to show how current changes versus radius.
5. What is the scalability of the system? In other words, what is the largest size of the sample that can trigger and maintain the oscillation?
6. The representation of Fig. 1h and 1i is confusing, as the results for other three trials have no axis labels.
7. Video S5, the length change of the microactuator is hard to observe.

Reviewer #2 (Remarks to the Author):

The authors present a well-studied engineering solution that takes benefit of periodically generated oxygen bubbles to power a micro-actuator.

The fact that bubbles under certain conditions arise periodically does not seem surprising to me, it has been studied multi-fold and even the evolution and rupture of bubbles is rather well modelled and understood.

The experiments are carefully planned, executed and all results are presented in a logic and well-designed manner.

The circuit and the microactuator powered by the bubbles are a nice integration, that shows that the potential of active matter has the potential to be explored on a larger scale.

Personally, I would have probably opted for a more specified journal, but the manuscript is very carefully drafted and I have no doubt about the manuscript quality.

Reviewer #3 (Remarks to the Author):

The core of this work reports about a very nice and simply designed experiment that includes two catalytically active particles that interact by their growing and bursting bubbles on the surface of an aqueous H₂O₂ solution. The interaction leads to an oscillatory behavior of the particles, which is studied as a function of number of particles and symmetry breaking. Stable oscillation is achieved in asymmetric multi-particle configurations. The behavior is nicely described and understood by theoretical modeling. Finally, the oscillation is exploited to generate oscillatory currents and voltages to drive an electrochemical microactuator. I think this is a very carefully executed study which is nice to read and understand. Still, there are some issues I would like the authors to consider:

(i) The underlying mechanism is the growth, coalescence and burst of microbubbles attached to one or more particles. Such systems are well-known in the literature and have been used to self-propel microtubular engines [Adv. Mater. 20, 4085 (2008); Small 5, 1688 (2009)]. These microengines themselves are self-oscillating autonomous systems with controllable oscillation frequencies. If gathered in a swarm [Adv. Mater. 22, 4340 (2010)] they show clear collective and stable oscillations [Nanoscale 5, 1284 (2013)] similar to the present study. Many prospective application and communication scenarios have been envisioned for collective bubble-microparticle swarms in the Nanoscale paper. I think the authors need to more carefully discuss existing literature in their manuscript and rephrase certain parts of their abstract/introduction and conclusion, e.g. "Here, we report the emergence of a low-frequency relaxation oscillator from a simple collective of active microparticles interacting at the air-liquid interface of a hydrogen peroxide drop"

(ii) The authors often refer to the beating robustness of their system – and under stable experimental conditions the system looks indeed impressively stable and robust. Still, I am wondering how the oscillatory behavior changes when the fluid is not kept still but undergoes fluid flow and/or turbulences in one or another way. Will this affect the robust behavior of the particle beating? And if yes, to what degree?

(iii) A somewhat weak point is the comparison to biological systems, which the authors feel they need to mimic at low-frequency oscillations. The authors for instance mention insect wings, but those operate at frequencies up to kHz – orders of magnitude higher than the oscillations demonstrated in the present study.

(iv) There is a somewhat strange paragraph in the introduction "Alternative mechanisms...energy modules.", which essentially says that slow oscillations have not been demonstrated in microscopic systems. However and apart from bubble propelled microengines (see (i)), this is exactly what was demonstrated in Ref. 22 ("Self-sufficient self-oscillating microsystem driven by low power at low Reynolds numbers") even in a biocompatible way. Hence, the authors should rephrase this part.

If the authors take care of these issues I am more than happy to consider recommending their work for publication in Nat. Comm.

Emergent Microrobotic Oscillators via Asymmetry-Induced Order

Response to Reviewer Comments

Jing Fan Yang* Thomas A. Berrueta* Allan M. Brooks Albert T. Liu
Ge Zhang David Gonzalez-Medrano Sungyun Yang Volodymyr Koman
Pavel Chvykov Lexy N. LeMar Marc Z. Miskin Todd D. Murphey
Michael S. Strano

Comments by Reviewer 1

The author reported an oscillatory microrobot fueled by hydrogen peroxide. The bubbling generation, merging with neighbor bubbles, bubble repulsion and particle attraction lead to a self-sustained oscillation system. The author studied multiple particle systems from experimental and theoretical pathways. The oscillatory motion could be used to trigger alternative electrical signals, which could power periodic motion of a micro-actuator. It is impressive that the authors were able to develop the microgenerator application from the oscillation phenomenon. Meanwhile, this method suffers from many intrinsic limitations (in terms of stability, long-term operation, etc.), though it is understandable for a proof-of-concept application. Overall, it is an interesting story with clear description. The reviewer recommends publishing the article in Nature Communication after addressing some comments below.

We thank the reviewer for recognizing our work's scientific values and appreciate the thoughtful feedback. We believe the reviewer will find the revised manuscript clarified and strengthened with additional experiments and calculations.

1. The author showed the tunability of the oscillation frequency. Is there any way to change amplitude, which is radius, in this system?

We explored the reviewer's suggestion with additional experiments. In the revised manuscript, we explored the dependence of the oscillation amplitude on (i) volume of the H_2O_2 drop on which the particles beat, and (ii) the size of the particles themselves.

First, as demonstrated and explained in the new Supplementary Fig. 7a within the Supplementary Information (copied below for the reviewer's convenience), the amplitude of oscillation (i.e. the maximum breathing radius of each cycle) increases with increasing volumes of H_2O_2 due to the reduced interfacial curvature around the drop apex. Shown in Supplementary Fig. 7b, the increased amplitude is accompanied by a longer period between consecutive beats, which is equivalent to a reduced oscillatory frequency.

Second, we explored the downward scalability of the collective oscillation with experiments of $250\mu\text{m}$ - and $100\mu\text{m}$ -diameter particles. We confirm that the collective oscillation observed in systems of two $500\mu\text{m}$ -particles does scale down to at least $100\mu\text{m}$, with changes in the oscillation amplitude and period as shown in the new Supplementary Fig. 8 (copied below). While the smaller particles still oscillate robustly, they

*These authors contributed equally.

do not travel as far (Supplementary Fig. 8a). This is primarily due to the reduced threshold bubble size, which translates to less impulse and kinetic energy conferred to the particles as they part. At the same time, the small particles beat at a lower frequency despite the shorter distance to cover (Supplementary Fig. 8b), because the bubble generation is slowed due to the reduced Pt areas.

Supplementary Figure 7: **Maximum breathing radius and interarrival time of two identical particles as a function of the H_2O_2 volume.** A larger volume of H_2O_2 solution corresponds to a reduced curvature of the liquid-air interface the particles reside in, which in turn weakens the global restorational force that resists parting of the particles. The breathing radius (a) therefore increases with the H_2O_2 volume, which consequently lengthens the intervals between consecutive bubble collapses (b). Due to the periodicity of all these 2-particle systems, the respective interarrival times are equivalent to the periods of oscillation. Each error bar denotes a standard deviation among all particle beats within a continuous experiment.

Supplementary Figure 8: **Maximum breathing radius and interarrival time of two identical particles as a function of the particle size.** All particles were fabricated by depositing 5nm Cr and 50nm Pt onto 10 μm -thick SU-8 polymer. The 500 μm , 250 μm , and 100 μm -diameter particles were designed to have Pt patches 250 μm , 125 μm , and 100 μm in diameter, respectively. Due to the periodicity of all these 2-particle systems, the respective interarrival times are equivalent to the periods of oscillation. Each error bar denotes a standard deviation among all particle beats within a continuous experiment.

In addition to the new supplementary figures, we have also modified the “Emergent low-frequency oscillation” segment of the main text to highlight these handles of tuning the oscillatory behaviour besides the H_2O_2 concentration:

“... the beating frequency’s dependence on H_2O_2 concentrations points to a mechanism for exerting fine control over the beating frequency, as predicted by our mechanistic model based on a Langmuir-Hinshelwood kinetics of the catalytic surface (Fig. 1j) [1, 2]. In Supplementary Figs. 7 and 8, we further explored the dependence of the oscillation amplitude and frequency on H_2O_2 volume and particle size. Of note, the stable emergent self-oscillation presented in Fig. 1 does scale down to $250\mu\text{m}$ - and $100\mu\text{m}$ -diameter particles.”

2. For the homogeneous system where relative burst intensity is 1, increasing the number of particles leads to loss of periodicity. However, why does increasing the number of particles not change \mathcal{R} ?

In Fig. 2d and Supplementary Fig. 2, we are interested in how particle heterogeneity brings about relative changes in the Rattling of each system. To facilitate a straightforward comparison of the relative lowering of \mathcal{R} as a function of N , we offset the curves to make all homogeneous systems share the same origin of $\mathcal{R} = 0$. We thank the reviewer for pointing this out in their review, and we have updated the captions of Fig. 2d to clarify our method of baselining:

“The curves are offset to make all $\mathcal{R} = 0$ at 1x intensity to highlight the effect of system heterogeneity on Rattling. See Supplementary Section 2 for a detailed discussion of the analytical model.”

Similarly, Supplementary Fig. 2 is updated accordingly:

“Note that we subtract the constant offset in rattling due to system size so that $\mathcal{R} = 0$ at $U = 0$ for all N .”

The newly added paragraph to the Supplementary Information (p.7) explains in full detail how the offset is calculated, as well as provides justification for it on the basis of Rattling Theory:

“Rearranging the expression in Supplementary Equation (18), we have the following expression

$$\mathcal{R}(U_{DL}, N) = -U_{DL} + \log \left(\frac{(N-1)e^{-(N-1)\bar{U}}}{(e^{-U_{DL}} + (N-1)e^{-\bar{U}})^N} \right),$$

which allows us to make predictions about the behavior of a collection of N beating particles with a single designated leader.

However, much in the same way that entropy can trivially depend on system size (e.g., number of microstates), our expression for rattling in Supplementary Equation (19) does as well. Thus, to focus on the dependence of $\mathcal{R}(U_{DL}, N)$ on U_{DL} , we subtract the constant bias that system size contributes to the value of rattling. To do this, we calculate $\mathcal{R}(U_{DL}, N) - \mathcal{R}(\bar{U}, N)$ for a choice of \bar{U} that we fix across all system sizes, where we note that $\mathcal{R}(\bar{U}, N)$ is merely a constant that offsets the value of rattling to be zero when $U_{DL} = \bar{U}$. Since $\mathcal{R}(\bar{U}, N)$ is exclusively a function of the number of particles for a given \bar{U} , subtracting it from $\mathcal{R}(U_{DL}, N)$ precisely removes the constant contribution of system size to the overall magnitude of rattling. As detailed in [3], constant offsets to the rattling values of a system do not affect its behavior. Only changes to the rattling landscape—that is, changes to the relative rattling values between configurations (or parameters)—have an effect on system behavior. This implies that comparing the absolute rattling values across systems is of limited use, which motivates our approach (as in Fig. 2d and Supplementary Fig. 2). ”

In summary, since Rattling captures the overall magnitude of system-level fluctuations, introducing additional degrees of freedom by increasing N introduces additional fluctuations, which trivially affects the absolute magnitude of \mathcal{R} at 1x. However, it is the *relative* difference in the Rattling of different configurations that determines a given system’s behavior (i.e., its “Rattling landscape”). By way of analogy, it is trivial that the absolute potential energy of a bag of five balls is larger than that of a single one at the same height. It is

only of interest to track how the potential energy of a given object changes relative to a given reference point. In our case, the reference point is set to be 1x intensity, to which we assign a \mathcal{R} of 0 for all N .

3. What is the energy efficiency of the microgenerator and mechanical oscillation? Will the generation of electrical signals decrease the mechanical oscillation’s amplitude?

In response to the reviewer’s thoughtful question, we added a new segment to the revised Supplementary Information (Section 5), which is reproduced below in full:

5 Note on the Energy Expenditure

5.1 Energy Conversions of the Mechanical Oscillation

Within each period of the emergent mechanical oscillation, chemical energy stored in the H_2O_2 fuel is converted into the particles’ kinetic energy upon the collapse of the O_2 bubble. The kinetic energy imparted to two outgoing particles simply take the form of $E_k = mv^2$, where m is the mass of each particle and v the maximal velocity right following the bubble collapse. With $m = 2.34\mu\text{g}$ for a $500\mu\text{m}$ -diameter particle and $v = 3.2 \times 10^4\mu\text{m/s}$ measured from experiments, E_{out} is estimated to be $2.40 \times 10^{-12}\text{J}$ per cycle. The chemical energy consumed per cycle may be computed as:

$$E_{\text{chem}} = \frac{2PV_{\text{b,th}}\Delta H}{RT}$$

where $V_{\text{b,th}}$ denotes the bubble volume at threshold, estimated to be $9.81 \times 10^{-2}\mu\text{L}$ in a 2-particle homogeneous system in 1mL of 10% H_2O_2 . We assume an ambient pressure P of 1atm and temperature T of 25°C , as the excess Laplace pressure within the bubble before collapse is a negligible $5.0 \times 10^{-3}\text{atm}$. ΔH , the enthalpy change of the decomposition reaction, is 98.24kJ/mol at given conditions, equivalent to an energy density of 2.89kJ/g H_2O_2 or 0.29kJ/g $10\text{wt}\%$ H_2O_2 solution [4]. E_{chem} per cycle is computed to be $7.88 \times 10^{-4}\text{J}$. The portion of the chemical energy converted to the work of expansion is:

$$W_{PV} = P_{\text{atm}}V_{\text{b,th}} + 4\pi\gamma R_{\text{b,th}}^2$$

where $R_{\text{b,th}}$ is the threshold radius assuming a spherical bubble. The latter term of $7.41 \times 10^{-8}\text{J}$ is the surface energy E_{surf} , i.e. the work against the Laplace pressure during bubble growth. To summarize, therefore, 1.26% of the original chemical energy contributes to a W_{PV} of $1.00 \times 10^{-5}\text{J}$. 0.74% of the work of expansion is stored as the surface energy. Finally, the kinetic energy gained by the particles account for 0.032% of surface energy stored in the bubble.

5.2 Energy Conversions of the Microgenerators

As the microgenerator converts the chemical energy from H_2O_2 decomposition to electrical work, it is of interest to calculate the proportion of total H_2O_2 molecules consumed which contributed to the electrical current [5, 6]. Given that each electrochemically redoxed H_2O_2 molecule transfers an electron, ON-state currents of 180.66nA (in the absence of an electrical load) and 15.27nA (with a load, i.e. the actuator) are respectively attributed to 1.87×10^{-12} and 1.58×10^{-13} moles of H_2O_2 per second. These correspond to 0.76% and 0.063% of the total peroxide consumption rate ($2P/RT \cdot dV_{\text{b}}/dt = 2.45 \times 10^{-9}\text{mol/s}$), respectively. The former is in agreement with prior literature [5], which estimated an electrochemical contribution of 0.5% . Since more than 99.9% of the consumed H_2O_2 decompose via the same non-electrochemical pathway as in the beating particles with no fuel cells aboard, generation of the electrical current has a negligible impact on the mechanical oscillation if all other conditions are kept the same. Along the same lines, additional fuel cell particles are not expected to diminish the electrical signals observed.

4. It is suggested to add the radius change over time in Figure 4e to show how current changes versus radius.

We have added Supplementary Fig. 16 below to the Supplementary Information. In this experiment, we simultaneously tracked the breathing radius of the particle system and measured the electrical current over time as the reviewer suggested:

Supplementary Figure 16: **Oscillatory mechanical beating drives on-board oscillatory current.** (See also Fig. 4e of the main text). As a standard 500- μm particle beats with a Pt-Ru fuel cell device (Fig. 4b, also Methods), the bubbles collapse at regular intervals as indicated by the spikes in the breathing radius trajectory ($r(t)$, top). Removal of the bubbles restores the electrochemical reactivity of the fuel cell electrodes, and therefore the current (bottom) peaks precisely as $r(t)$ does. The current measured in this experiment is an order of magnitude higher than that in Fig. 4e since the system characterized here was not connected to an actuator.

As the imaging of the particles and the actuators require distinct optical focus and zoom conditions, it is challenging with our experimental setup to simultaneously record the breathing radius and the actuation. Nevertheless, the new Supplementary Fig. 16 above, when considered in conjunction with Fig. 4e of the main text, provides conclusive evidence that the emergent particle beating induces a periodic electrical current, which then drives the cyclical actuation. Figure 4e is copied below for a direct comparison:

Figure 4e: Cyclic motion of a microbotic actuator driven by the oscillatory current. The current measurement over time and the actuator length change closely match, confirming that the cyclic actuation is driven by the oscillatory current, which itself is emergent from the particle beating.

5. What is the scalability of the system? In other words, what is the largest size of the sample that can trigger and maintain the oscillation?

The scale the reviewer enquired may be interpreted as either the number of particles (N) in a collective or the size of each individual particle. We have discussed the former in the context of periodic beating in the manuscript. As presented in Fig. 2b of the main text, periodicity of the emergent self-oscillation breaks down as N increases in homogeneous systems. Beyond $N = 7$, their interarrival distributions become statistically indistinguishable from those of a Poisson process. In contrast, heterogeneous systems with a broken permutation symmetry maintains their periodicity across $N = 2$ to 11, evident from Figs. 2e, 2g, as well as Supplementary Figs. 10 and 11. As seen in the micrographs of Supplementary Fig. 11, particles in $N = 10$ and 11 heterogeneous systems start to deplete positions immediately adjacent to the DL even at the threshold bubble size. It is therefore reasonable to expect a certain degree of particle motion arising from interactions among distant particles that are unable to access the DL’s “sphere of influence.” These irregularities in the signal are however expected to be minor in contrast to the collapse events of the DL’s central bubble, which broadly impact all particles in the neighbourhood.

While periodic self-oscillation is not expected from a large-scale homogeneous collective, we do find intriguing the new physics of hierarchical bubble organization in our experiment with 50 identical particles, which we intend to investigate further in our future work:

Supplementary Figure 12: Progression of a large-scale homogeneous collection. While we have established in Fig. 2 of the main text that the periodicity of homogeneous systems breaks down easily as N increases, we observe intriguing hierarchical organization of the bubbles in this 50-particle collective over a span of 8s. Bubbles from individual particles merge and grow (i), resulting in the intermediate situation in (ii) where a large bubble situated at the H_2O_2 drop’s apex is packed around by smaller ones. Following further merger and growth (iii), the system eventually collapses (iv). Highlighted in yellow circles are bubbles larger than $350\mu\text{m}$ in radius. With a number of particles distributed along the perimeter, a bubble is observed to grow far beyond the typical threshold size in few-particle homogeneous systems (*cf.* bubble sizes in DL systems in Supplementary Figs. 10 and 11). Scale bar, 1mm. All experiments were performed in 1mL of 10.7wt% H_2O_2 .

On the other hand, if we are to interpret “scalability” as that of the particle dimension, we have included in the revised manuscript new experiments with particles $250\mu\text{m}$ and $100\mu\text{m}$ in diameter. The emergent self-oscillation observed with $500\mu\text{m}$ particles still stands when their dimensions are scaled down (Supplementary Fig. 8, copied below). A more thorough discussion on the oscillation amplitude and frequency can be found in our response to the reviewer’s first comment. We did not explore particles larger than $500\mu\text{m}$ as they depart from the relevant length scales of microrobotics; particles smaller than $100\mu\text{m}$ are well below the size resolvable by the naked eye and are challenging to manipulate and transfer.

Supplementary Figure 8: **Maximum breathing radius and interarrival time of two identical particles as a function of the particle size.** All particles were fabricated by depositing 5nm Cr and 50nm Pt onto 10 μm -thick SU-8 polymer. The 500 μm , 250 μm , and 100 μm -diameter particles were designed to have Pt patches 250 μm , 125 μm , and 100 μm in diameter, respectively. Due to the periodicity of all these 2-particle systems, the respective interarrival times are equivalent to the periods of oscillation. Each error bar denotes a standard deviation among the oscillation cycles within an experiment.

6. The representation of Fig. 1h and 1i is confusing, as the results for other three trials have no axis labels.

We thank the reviewer for pointing out the non-intuitive way we visualized our data in the original submission. Through presenting results from 4 independent experiments we intended to convey the idea of reliability of the emergent self-oscillation behaviour. In the revised Figs. 1h and i we have made it clear that all subplots share the same axes, which is emphasized in the caption as well:

Revised Figure 1h,i: **h**, The phase portraits of 4 independent 2-particle experiments demonstrate reproducible limit cycles with closed-loop orbits, confirming the periodicity of collective beating. Note that to calculate the phase portraits the system's bubble-driven discontinuities were processed through a standard finite-impulse response filter (see Methods). All phase portraits share the same axes. **i**, The recurrence histograms of the same 4 experiments all display a narrow peak centred at a period of 3.2s, consistent with visual evidence in (e). All histograms share the same axes.

Original Figure 1h,i: ...

7. Video S5, the length change of the microactuator is hard to observe.

In the edited Supplementary Movie 5, we accentuated the previously subtle motion of the micro-actuator with the following changes:

- We accelerated the original movie to a playback speed found to be optimal for observing the micro-actuator changes;
- We flat-field corrected the movie to remove the dark shaded region at the bottom-left of the original video, which may have been a distracting factor. The region with the micro-actuator was minimally affected;

- We learned that the cyclic motion is best observed if the viewer focuses his/her attention at the tip of the micro-actuator. We included the instruction card reproduced below (left) in the movie before it starts playing;
- We appended a second video clip to the original movie (see the intertitle reproduced below, on the right). This clip loops through 1.5s of the original movie, during which the actuator is observed to extend and contract. The rapid succession of the actuator cycles should make the length change apparent to the viewer.

Supplementary Movie 5 of 5:
Cyclic actuation of a microrobotic arm, driven by the on-board oscillatory electrical current

Viewer of the video is advised to focus on the tip of the micro-actuator to best observe its cyclic motion.

The video is sped up 35x.

Supplementary Movie 5 of 5:
Cyclic actuation of a microrobotic arm, driven by the on-board oscillatory electrical current

The next part of the video loops through a 1.5s electrical current pulse. The extension and contraction of the micro-actuator during the period is easily observable.

The video is sped up 35x.

Supplementary Movie 5: **The cyclic actuation of a microrobotic arm driven by the on-board oscillatory electrical current.** Illustrated in Fig. 4a, we translated the chemomechanical beating to a usable periodic electrical signal via a simple on-board bimetallic fuel cell. This current actuates the tip of the microrobotic arm repeatedly with minimal phase delay, shown in the time series data of the measured current (I) and the actuator length change (ΔLength). **The video was sharpened with the unsharp masking technique and flat-field corrected. It was sped up 35 times.**

Comments by Reviewer 2

The authors present a well-studied engineering solution that takes benefit of periodically generated oxygen bubbles to power a micro-actuator. The fact that bubbles under certain conditions arise periodically does not seem surprising to me, it has been studied multi-fold and even the evolution and rupture of bubbles is rather well modelled and understood. The experiments are carefully planned, executed and all results are presented in a logic and well-designed manner. The circuit and the microactuator powered by the bubbles are a nice integration, that shows that the potential of active matter has the potential to be explored on a larger scale. Personally, I would have probably opted for a more specified journal, but the manuscript is very carefully drafted and I have no doubt about the manuscript quality.

We thank the reviewer for recognizing the quality of our manuscript and of our research, particularly highlighting the practical impacts of the on-board electrical signal transduction. We are aware of a rich body of literature on the physics of bubbles and in fact constructed our mechanistic model based on these works (as well as provided appropriate citations to them in the text, e.g., [7–9]). In addition to our experimental contributions, we believe that our work presents substantial advances to the theory of nonequilibrium thermodynamics in active matter. Our study surrounding the thermodynamic mechanisms underlying the emergence (and stability) of our microparticle oscillators, as well as our analyses on the relationship between permutation symmetry-breaking and collective order, are far-from-trivial discoveries with implications in the physics of self-organization. Because of the potential interest to engineers and physicists (and experimentalists and theorists) alike, we do believe an interdisciplinary journal such as *Nature Communications* is a suitable home for our manuscript.

Comments by Reviewer 3

The core of this work reports about a very nice and simply designed experiment that includes two catalytically active particles that interact by their growing and bursting bubbles on the surface of an aqueous H_2O_2 solution. The interaction leads to an oscillatory behavior of the particles, which is studied as a function of number of particles and symmetry breaking. Stable oscillation is achieved in asymmetric multi-particle configurations. The behavior is nicely described and understood by theoretical modeling. Finally, the oscillation is exploited to generate oscillatory currents and voltages to drive an electrochemical microactuator. I think this is a very carefully executed study which is nice to read and understand. Still, there are some issues I would like the authors to consider.

We would like to thank the reviewer for taking the time to develop a detailed understanding of our experimental results and for regarding highly the quality of our manuscript. The reviewer's comments have been very useful towards improving the clarity and framing of our contributions.

(i) The underlying mechanism is the growth, coalescence and burst of microbubbles attached to one or more particles. Such systems are well-known in the literature and have been used to self-propel microtubular engines [Adv. Mater. 20, 4085 (2008); Small 5, 1688 (2009)]. These microengines themselves are self-oscillating autonomous systems with controllable oscillation frequencies. If gathered in a swarm [Adv. Mater. 22, 4340 (2010)] they show clear collective and stable oscillations [Nanoscale 5, 1284 (2013)] similar to the present study. Many prospective application and communication scenarios have been envisioned for collective bubble-microparticle swarms in the Nanoscale paper. I think the authors need to more carefully discuss existing literature in their manuscript and rephrase certain parts of their abstract/introduction and conclusion, e.g. ‘‘Here, we report the emergence of a low-frequency relaxation oscillator from a simple collective of active microparticles interacting at the air-liquid interface of a hydrogen peroxide drop’’

We have made several changes to our manuscript that address the reviewer's comments in an effort to better situate the novelty of our work within pre-existing literature, such as the ones highlighted by the reviewer.

First, we have rephrased our contributions in the abstract, introduction, and discussion to avoid potential ambiguity on our scientific contributions. Our claim to novelty is the use of—and analytical derivation of mechanisms for—self-organization to produce robust and useful mechanical and electrical oscillations in microparticle collectives. As such, we have altered our potentially ambiguous language in multiple places to reflect our emphasis. In the abstract we modified the sentence highlighted by the reviewer to prevent the words ‘‘report’’ and ‘‘emergent’’ from implying that our central finding is the experimental observation of a mechanical oscillation:

‘‘Here, we study a low-frequency relaxation oscillator that emerges from the interactions of a simple collective of active microparticles at the air-liquid interface of a hydrogen peroxide drop.’’

Additionally, we have added the words ‘‘the discovery of a thermodynamic mechanism’’ to the following sentence in the abstract to emphasize that an important portion of our novelty comes from our theoretical contributions:

‘‘We explain such emergent order through the discovery of a thermodynamic mechanism for asymmetry-induced order.’’

In the last paragraph of the introduction, we have also rephrased our first sentence to be clearer and much more direct in summarizing our results:

“In this work, instead of relying on complex chemistries, integrated electronics, or elaborate mechanical microstructures, we produce robust electromechanical oscillations aboard a collective of deceptively simple microparticles by exploiting the self-organized properties of their far-from-equilibrium dynamics.”

Finally, in the discussion section we now cite the reviewer’s suggested reference (i.e., Solovev *et al.* [10]), which indeed contains valuable and forward-thinking designs, in our treatment of future physical communication modes among particles:

“The use of on-board electrical currents will enable the integration of sensors and computational elements to enrich physical microparticle interactions [–], forming the basis for future collectives wherein the long-envisioned potential of complex inter-particle communications can be implemented [10].”

Second, we thank the reviewer for highlighting the literature on large collectives of bubble-propelled tubular microswimmers, which were observed to dynamically self-assemble at the air-liquid interface ([10] and [11]). We do see a connection between these references and our discussion of the phenomenology of our system in terms of the interfacial forces and catalytic bubble generation. We have now highlighted this connection in the second paragraph of the “Emergent low-frequency oscillation” subsection of the Results. In this same paragraph, we have added a mention of the other suggested references on bubble-propelled microtubules (i.e., [12] and [13]) as well.

“The collapse imparts an impulse onto the microparticles and propels them in opposite directions, at which point the particles are drawn back towards one another by the restorational forces: First, the radial component of buoyancy, \mathbf{F}_g , globally directs the particles towards the apex of the concave air-liquid interface [–]. Second, the local interfacial deformations result in a mutual attractive capillary force \mathbf{F}_c , affectionately known as the “Cheerios effect” [–,–]. **The combination of this Cheerios effect and catalytic bubble generation has been observed to produce repetitive back-and-forth motion [10, 11] in swarms of tubular swimmers [12, 13].**”

Lastly, while we relate to the reviewer’s intuition in discussing periodic bubble generation in catalytic microtubules as self-oscillating systems, our work focuses on the periodic movement of the microparticles themselves—rather than the periodicity of the forcing provided by the fuel source—in line with both theoretical formulations [14] and experimental realizations [15–18] of self-oscillation.

(ii) The authors often refer to the beating robustness of their system -- and under stable experimental conditions the system looks indeed impressively stable and robust. Still, I am wondering how the oscillatory behavior changes when the fluid is not kept still but undergoes fluid flow and/or turbulences in one or another way. Will this affect the robust behavior of the particle beating? And if yes, to what degree?

We are pleased the reviewer finds our results impressive. In the revised manuscript, we included additional experiments (the new Supplementary Fig. 9 copied below) which shows how a two-particle system quickly resumes periodic beating following various forms of disturbances, with the oscillation amplitude and periodicity unchanged. These results further corroborate the robustness of our system in the context of dynamical systems, defined as the resilience to perturbations, or equivalently, the ability to revert back to the prior stable configuration [19]. Deforming the liquid-air interface with a pipette as a form of perturbation is inspired by Solovev *et al.* recommended by the reviewer [11].

Supplementary Figure 9: **Robustness of the emergent oscillation to perturbations.** In these two experiments, we intentionally disturbed a system of two identical particles by (i) deforming the liquid-air interface with a pipette [11], (ii) stirring the H_2O_2 drop, and (iii) shaking the drop back and forth. It is evident in the breathing radius trajectories that the collective oscillation resumes promptly following the perturbations (shaded region) with its amplitude and periodicity unchanged, thus demonstrating robustness. Data discontinuities during the perturbations are a result of blurry frames or particles temporarily exiting the camera field-of-view. The inset micrograph shows the particles approaching the pipette due to the deformed interface. Scale bar, 1mm.

Additionally, the “Emergent low-frequency oscillation” section of the main text has been updated in light of our new results:

“The period remains constant throughout as revealed by the moving-window recurrence analyses (Supplementary Fig. 6, Methods), since a negligible 0.02% of the fuel is consumed over 280s based on stoichiometry. Furthermore, the oscillation amplitude and periodicity are shown to be resilient towards various forms of perturbations (Supplementary Fig. 9). We developed a mechanistic model based on ...”

(iii) A somewhat weak point is the comparison to biological systems, which the authors feel they need to mimic at low-frequency oscillations. The authors for instance mention insect wings, but those operate at frequencies up to kHz -- orders of magnitude higher than the oscillations demonstrated in the present study.

In our parallel to biological systems, which only appears briefly at the very beginning of the introduction, we merely hoped to provide the reader with motivation for the general importance of low-frequency oscillations in nature. We never intended to suggest (nor claimed in the text) that the frequencies of our oscillators cover the entire range of those observed in nature, nor did we claim that our oscillatory mechanism is biomimetic. We believe our text is clear on these points; therefore, we respectfully argue that readers will not be misled by our use of biological motivation.

We believe the reviewer would agree that parallels to biological systems provide valuable intuition to better understand research on low-frequency oscillators, as well as other topics—three of the four citations highlighted by the reviewer also have parallels to biological systems in their introductory sections [10, 11, 13]. Low-frequency oscillations are essential to the autonomy of biological systems because they help to coordinate the complex interactions of many simplistic elements. Heartbeats [20], neuronal activity [21], locomotion [22], breathing [23], sleep [24], and immunity [25] are just a few examples of behaviors critical to organism autonomy enabled by low-frequency oscillations. As a result of their broad importance in biology, engineers have explored the roles that low-frequency oscillations can play in artificial autonomous systems, and have found them to be crucial to the development of novel functionalities in macroscopic robots [26], soft robots [4], and microrobots [27].

We thank the reviewer for bringing up the frequency of the flaps of insect wings, which prompted us to perform a more thorough literature research on the topic. Intriguingly, oscillations in the kHz range are only possible in insects with asynchronous wing movement [28], where phase offsets between wings artificially create a wingbeat frequency that is faster overall. In contrast, synchronous insect wingbeats rarely exceed 100-200Hz. That being said, we did change our cited example from insect flight to horse galloping. This new framing should be more intuitive to a broad scientific audience and avoid misunderstandings.

“While complex electronics operate at ever-increasing clock rates of many gigahertz, the frequency of many important biological oscillations seldom exceeds 100Hz. The slow rate of these oscillations stems from a need to be commensurate with both the energy budget and the natural timescales of underlying biological processes, as in the transport of CO₂ in plants [–] and in the galloping of horses [29].”

(iv) There is a somewhat strange paragraph in the introduction ‘‘Alternative mechanisms. . . energy modules.’’, which essentially says that slow oscillations have not been demonstrated in microscopic systems. However and apart from bubble propelled microengines (see (i)), this is exactly what was demonstrated in Ref. 22 (‘‘Self-sufficient self-oscillating microsystem driven by low power at low Reynolds numbers’’) even in a biocompatible way. Hence, the authors should rephrase this part.

In our previous draft, the first paragraph (part of which is referenced by the reviewer) sought to communicate several points—potentially too many to present a clear framing. In response to the reviewer’s comment, we have split our first paragraph into two in hopes of improving the clarity of our ideas.

The text referenced by the reviewer, which is now part of the second paragraph of the new manuscript, sought to communicate the challenges and progress towards producing low-frequency oscillations in artificial microsystems. We agree with the reviewer that Akbar *et al.* [27], as a major step towards untethered microscale oscillators, should be explicitly discussed. We are particularly fond of Akbar *et al.* also because the modulation of electrical resistance with mechanical oscillation is analogous to our mechanism of generating on-board oscillatory currents. We have modified the entire paragraph in response to the reviewer’s comment and provided special mention to the work of [27].

“In artificial microsystems, however, the production of slow self-sufficient self-oscillations is counterintuitively difficult [-,-]. Generating self-sustaining *mechanical* oscillations at the microscale typically requires the transduction of complex chemical oscillators (e.g., Belousov-Zhabotinsky reaction [-]) into periodic changes to a system’s physical configuration [-,-,-,-,-,-]. Alternative mechanisms for producing self-sufficient mechanical oscillations based on carefully designed dynamic coupling between responsive elastic materials and thermal [-,-], chemical [-,-,-], or moisture stimuli [-] have typically been demonstrated in millimetre-scale (and larger) devices. In contrast, generating slow periodic *electrical* signals remains prohibitively challenging aboard untethered microscale devices (Supplementary Section 3), given the limited downward scalability of capacitors and inductors [-,-], as well as the power and footprint demands of CMOS oscillators, frequency dividers, and energy modules [-,-,-]. Despite these challenges, recent progress has shown that self-sustaining electrical oscillations can be produced by modulating electrical resistance with mechanical feedback loops in carefully designed devices, presenting a promising mechanism for sub-500 μm electrical self-oscillators [27].”

Moreover, we have refined our statement of contributions (first sentence of the final paragraph of the introduction) to reflect the ways in which our work is distinct from [27]. Namely, that our work seeks to take advantage of self-organization and emergent nonequilibrium order in the pursuit of low-frequency electromechanical self-oscillations.

“In this work, instead of relying on complex chemistries, integrated electronics, or elaborate mechanical microstructures, we produce robust electromechanical oscillations aboard a collective of deceptively simple microparticles by exploiting the self-organized properties of their far-from-equilibrium dynamics.”

Additionally, we would like to point out the much smaller scale of our systems in contrast to the sophisticated devices in [27], which exceed 500 μm even if we do not consider the mm-scale power supply required for untethered operation. In the revised manuscript, we have included additional data of scaled down microparticles 250 μm and 100 μm in diameter, which still showed robust low-frequency oscillation. The error bars reveal the minimal variability :

Supplementary Figure 8: **Maximum breathing radius and interarrival time of two identical particles as a function of the particle size.** All particles were fabricated by depositing 5nm Cr and 50nm Pt onto 10 μm -thick SU-8 polymer. The 500 μm , 250 μm , and 100 μm -diameter particles were designed to have Pt patches 250 μm , 125 μm , and 100 μm in diameter, respectively. Due to the periodicity of all these 2-particle systems, the respective interarrival times are equivalent to the periods of oscillation. Each error bar denotes a standard deviation among the oscillation cycles within an experiment.

If the authors take care of these issues I am more than happy to consider recommending their work for publication in Nat. Comm.

We thank the reviewer again for their diligence and appreciation for the quality of our work. We believe their comments have greatly benefited our manuscript.

References

1. Lin, S.-S. & Gurol, M. D. Catalytic Decomposition of Hydrogen Peroxide on Iron Oxide: Kinetics, Mechanism, and Implications. *Environmental Science & Technology* **32**, 1417–1423. ISSN: 0013-936X (1998).
2. Plauck, A., Stangland, E. E., Dumesic, J. A. & Mavrikakis, M. Active sites and mechanisms for H₂O₂ decomposition over Pd catalysts. *Proceedings of the National Academy of Sciences* **113**, E1973–E1982 (2016).
3. Chvykov, P. *et al.* Low rattling: A predictive principle for self-organization in active collectives. *Science* **371**, 90–95. ISSN: 0036-8075 (2021).
4. Wehner, M. *et al.* An integrated design and fabrication strategy for entirely soft, autonomous robots. *Nature* **536**, 451–455 (2016).
5. Wang, W., Chiang, T.-y., Velegol, D. & Mallouk, T. E. Understanding the efficiency of autonomous nano- and microscale motors. *Journal of the American Chemical Society* **135**, 10557–10565 (July 2013).
6. Paxton, W. F. *et al.* Catalytically Induced Electrokinetics for Motors and Micropumps. *Journal of the American Chemical Society* **128**, 14881–14888 (2006).
7. Chen, S.-l., Lin, C.-t., Pan, C., Chieng, C.-c. & Tseng, F.-g. Growth and detachment of chemical reaction generated micro-bubbles on micro-textured catalyst. *Microfluidics and Nanofluidics* **7**, 807. ISSN: 1613-4990 (2009).
8. Moreno Soto, A., Maddalena, T., Fraters, A., van der Meer, D. & Lohse, D. Coalescence of diffusively growing gas bubbles. *Journal of Fluid Mechanics* **846**, 143–165 (2018).
9. Lv, P. *et al.* Self-Propelled Detachment upon Coalescence of Surface Bubbles. *Phys. Rev. Lett.* **127**, 235501 (23 2021).
10. Solovev, A. A., Sanchez, S. & Schmidt, O. G. Collective behaviour of self-propelled catalytic micromotors. *Nanoscale* **5**, 1284–1293 (4 2013).
11. Solovev, A. A., Mei, Y. & Schmidt, O. G. Catalytic Microstrider at the Air–Liquid Interface. *Advanced Materials* **22**, 4340–4344 (2010).
12. Mei, Y. *et al.* Versatile Approach for Integrative and Functionalized Tubes by Strain Engineering of Nanomembranes on Polymers. *Advanced Materials* **20**, 4085–4090 (2008).
13. Solovev, A. A., Mei, Y., Bermúdez Ureña, E., Huang, G. & Schmidt, O. G. Catalytic Microtubular Jet Engines Self-Propelled by Accumulated Gas Bubbles. *Small* **5**, 1688–1692 (2009).
14. Jenkins, A. Self-oscillation. *Physics Reports* **525**, 167–222. ISSN: 0370-1573 (2013).
15. Maeda, S., Hara, Y., Sakai, T., Yoshida, R. & Hashimoto, S. Self-Walking Gel. *Advanced Materials* **19**, 3480–3484 (2007).
16. Hua, M. *et al.* Swaying gel: Chemo-mechanical self-oscillation based on dynamic buckling. *Matter* **4**, 1029–1041. ISSN: 2590-2385 (2021).
17. Yoshida, R. Self-Oscillating Gels Driven by the Belousov–Zhabotinsky Reaction as Novel Smart Materials. *Advanced Materials* **22**, 3463–3483 (2010).
18. Zhao, Y. *et al.* Soft phototactic swimmer based on self-sustained hydrogel oscillator. *Science Robotics* **4**, eafax7112 (2019).
19. Sidorov, E. & Zacksenhouse, M. Lyapunov based estimation of the basin of attraction of Poincare maps with applications to limit cycle walking. *Nonlinear Analysis: Hybrid Systems* **33**, 179–194. ISSN: 1751570X. <https://doi.org/10.1016/j.nahs.2019.03.002> <https://linkinghub.elsevier.com/retrieve/pii/S1751570X19300330> (Aug. 2019).
20. Madwed, J. B., Albrecht, P., Mark, R. G. & Cohen, R. J. Low-frequency oscillations in arterial pressure and heart rate: A simple computer model. *American Journal of Physiology-Heart and Circulatory Physiology* **256**, 1573–1579 (1989).
21. Buzsáki, G. & Draguhn, A. Neuronal Oscillations in Cortical Networks. *Science* **304**, 1926–1929 (2004).

22. Guertin, P. A. The mammalian central pattern generator for locomotion. *Brain Research Reviews* **62**, 45–56. ISSN: 0165-0173 (2009).
23. Von Euler, C. On the central pattern generator for the basic breathing rhythmicity. *Journal of Applied Physiology* **55**, 1647–1659 (1983).
24. Whitmore, D., Foulkes, N. S. & Sassone-Corsi, P. Light acts directly on organs and cells in culture to set the vertebrate circadian clock. *Nature* **404**, 87–91 (2000).
25. Wang, C., Lutes, L. K., Barnoud, C. & Scheiermann, C. The circadian immune system. *Science Immunology* **7**, eabm2465 (2022).
26. Thor, M. & Manoongpong, P. Versatile modular neural locomotion control with fast learning. *Nature Machine Intelligence* **4**, 169–179. ISSN: 2522-5839 (Feb. 2022).
27. Akbar, F. *et al.* Self-sufficient self-oscillating microsystem driven by low power at low Reynolds numbers. *Science Advances* **7**, eabj0767 (2021).
28. Josephson, R., Malamud, J. & Stokes, D. Asynchronous muscle: A primer. *Journal of Experimental Biology* **203**, 2713–2722. ISSN: 0022-0949 (Sept. 2000).
29. Hoyt, D. F. & Taylor, C. R. Gait and the energetics of locomotion in horses. *Nature* **292**, 239–240. ISSN: 1476-4687 (July 1981).

REVIEWERS' COMMENTS

Reviewer #1 (Remarks to the Author):

All my comments have been properly addressed by the authors. I do appreciate their effort to explore questions that were beyond the scope of the initial manuscript, including the tunability of oscillation amplitude, the scalability of the system, etc. In all, I think the quality of the manuscript has been sufficiently improved and I recommend publishing it on Nature Communications.

Reviewer #3 (Remarks to the Author):

The authors have addressed my comments very well. I recommend publication of the manuscript in Nat. Comm.

Manuscript NCOMMS-22-19130A

Emergent Microrobotic Oscillators via Asymmetry-Induced Order

Comments from REVIEWER 1

All my comments have been properly addressed by the authors. I do appreciate their effort to explore questions that were beyond the scope of the initial manuscript, including the tunability of oscillation amplitude, the scalability of the system, etc. In all, I think the quality of the manuscript has been sufficiently improved and I recommend publishing it on *Nature Communications*.

Reply:

We thank the reviewer for the thoughtful feedback which had improved our manuscript. We also appreciate their acknowledgement of our additional experiments and analyses added during the review process.

Comments from REVIEWER 3

The authors have addressed my comments very well. I recommend publication of the manuscript in *Nat. Comm.*

Reply:

We are delighted with the reviewer's approval and appreciate their feedback and suggestions during review.